# Prospect of Plasmacytoid Dendritic Cells in Enhancing Anti-Tumor Immunity of Oncolytic Herpes Viruses

**DOI:** 10.3390/cancers11050651

**Published:** 2019-05-11

**Authors:** Philipp Schuster, Georg Lindner, Sabrina Thomann, Sebastian Haferkamp, Barbara Schmidt

**Affiliations:** 1Institute of Medical Microbiology and Hygiene, University of Regensburg, 93053 Regensburg, Germany; philipp1.schuster@ukr.de (P.S.); georg1.lindner@ukr.de (G.L.); 2Institute of Clinical and Molecular Virology, Friedrich-Alexander-Universität Erlangen-Nürnberg, 91054 Erlangen, Germany; sabrina.thomann@arcor.de; 3Department of Dermatology, University Medical Center, 93053 Regensburg, Germany; sebastian.haferkamp@ukr.de; 4Institute of Clinical Microbiology and Hygiene, University Medical Center, 93053 Regensburg, Germany

**Keywords:** plasmacytoid dendritic cells, herpes simplex virus 1, oncolytic virus

## Abstract

The major type I interferon-producing plasmacytoid dendritic cells (pDC) surround and infiltrate certain tumors like malignant melanoma, head and neck cancer, and ovarian and breast cancer. The presence of pDC in these tumors is associated with an unfavorable prognosis for the patients as long as these cells are unstimulated. Upon activation by synthetic Toll-like receptor agonists or viruses, however, pDC develop cytotoxic activities. Viruses have the additional advantage to augment cytotoxic activities of pDC via lytic replication in malignant lesions. These effects turn cold tumors into hotspots, recruiting further immune cells to the site of inflammation. Activated pDC contribute to cross-presentation of tumor-associated antigens by classical dendritic cells, which induce cytotoxic T-cells in particular in the presence of checkpoint inhibitors. The modification of oncolytic herpes viruses via genetic engineering favorably affects this process through the enhanced production of pro-inflammatory cytokines, curbing of tumor blood supply, and removal of extracellular barriers for efficient viral spread. Importantly, viral vectors may contribute to stimulation of memory-type adaptive immune responses through presentation of tumor-related neo- and/or self-antigens. Eventually, both replication-competent and replication-deficient herpes simplex virus 1 (HSV-1) may serve as vaccine vectors, which contribute to tumor regression by the stimulation of pDC and other dendritic cells in adjuvant and neo-adjuvant situations.

## 1. Introduction

Based on early observations of lineage-negative cells with dendritic cell (DC)-like morphology [1,2], plasmacytoid dendritic cells (pDC) were characterized by two independent groups in 1999 as the main type I interferon (IFN)-producing cells in the blood upon stimulation with herpes simplex or influenza viruses [3,4]. Since then, it has become clear that these cells play an important role not only in innate and adaptive immune defenses against viruses and other pathogens but also in autoimmune diseases and anti-tumor immunity. Recent evidence suggests that pDC are a heterogeneous cell population consisting of a majority of “conventional” pDC and a minority of “pDC-like cells” originating from common lymphoid and common dendritic cell (DC) precursors, respectively [5]. The latter excel in antigen processing and presentation and may thus contribute to inducing antitumor responses in vivo [5].

Several excellent reviews contributed by well-known experts in the field address antiviral and anti-tumor activities of pDC [6,7], as well as properties of oncolytic viruses [8,9,10,11]. The purpose of our review is not to repeat these findings but to specifically focus on the role of pDC in anti-tumor defenses in the context of oncolytic herpes simplex virus 1 (HSV-1). In particular, we will address how the genetic engineering of oncolytic HSV-1 can contribute to targeted stimulation of pDC and other DC and thus contribute to tumor regression.

## 2. Facts and Prospects

### 2.1. The Yin and Yang of pDC within Tumors

pDC surround and occasionally infiltrate primary melanoma lesions and can also be detected in T-cell rich areas of sentinel lymph nodes in patients suffering from malignant melanoma [12,13,14]. In addition, pDC are reported to populate head and neck squamous cell tumors [15], ovarian carcinoma [16,17], and breast cancer [18,19]. The presence of immature pDC in these tumors is regularly associated with an immunosuppressive microenvironment, promoting regulatory immunity, and favoring tumor progression [12,20,21,22]. Specifically, tumor-infiltrating pDC were impaired in responding to Toll-like receptor (TLR) 9 but not TLR7 agonists [23].

Upon activation of pDC using potent stimuli, they start to exert anti-tumorigenic activity. In this activation, natural and synthetic TLR agonists play a major role. Thus, the accumulation of pDC and regression of murine malignant melanoma is reported upon stimulation with TLR7 agonist imiquimod [24] and TLR9 agonist CpG-B [25]. Activated pDC exert cytotoxic activity, mostly through a TRAIL-dependent mechanism [26,27], and stimulate other immune cells like CD4+ and CD8+ T-cells, as well as natural killer (NK) cells [28,29,30]. Most importantly, pDC cooperate with classical dendritic cells from the myeloid lineage in the anti-tumor defense [31]. In this process, cDC1 cells, although rare in tumors, have proven to be crucial for stimulation of CD8+ T cells via trafficking of tumor antigen to lymphatic tissue [32,33]. The recruitment of cDC1 cells into tumor tissue is dependent on natural killer cells [34,35].

In principle, pDC are in a dormant stage in the vicinity of malignant melanoma lesions but, upon proper activation, may attack tumor cells via direct killer cell-like cytotoxic activity and/or may induce systemic adaptive immune responses against tumor-specific antigens. 

### 2.2. The Viral Wake-Up Call 

Apart from synthetic TLR ligands, pDC can efficiently be activated using RNA and DNA viruses (Figure 1). In this respect, the human leukemic pDC cell line Gen2.2 developed cytotoxic activity against tumor cells upon stimulation with inactivated influenza virus [36]. Similar effects were observed with this cell line upon exposure to influenza virus-like virosomes delivering tumor peptides [37]. TRAIL-mediated cytotoxicity by human pDC was observed in vitro upon stimulation with live-attenuated measles virus vaccine [38]. Recently, the recruitment of pDC to mouse melanoma lesions was reported upon exposure to human ß-defensin-expressing vaccinia virus [39].

In a systematic approach, the Figdor group characterized the potential of prophylactic live-attenuated and inactivated viral vaccines to induce pDC activation and maturation, MHC class I and class II expression, IFN-α production, and T-cell proliferation [40]. These data promoted a first-in-patient trial in which pDC derived from patients suffering from metastasized malignant melanoma were loaded ex vivo with tumor peptides, activated using tick-borne encephalitis virus vaccine, and then re-infused into the patients, which significantly prolonged their overall survival [41]. This success pioneered the application of viruses as an efficient anti-tumor approach.

pDC can also be efficiently stimulated using HSV-1. HSV-1 is an enveloped double-stranded DNA virus of the subfamily Alphaherpesvirinae which lytically replicates in many infected cell types and establishes latent infection in sensory ganglions [42]. We have shown that wild type HSV-1 induces a timely coordinated regulation of pDC surface receptors involved in chemotaxis, maturation, migration, cytotoxicity, and costimulation [43], pointing to strong antiviral activities of pDC in HSV-1 infections [44]. These effects on activation, maturation, and migration markers were similar to those induced by different classes of CpG oligonucleotides [45] and cellular DNA species [46]. A subpopulation of pDC started to express CD8α upon exposure to HSV-1, displaying a highly activated phenotype, which may be particularly active in recruiting other immune cells via the secretion of IL-8, MIP-1alpha, MIP-1beta, and MCP-1 chemokines [47]. Furthermore, human NK cells were efficiently activated by IFN-α and TNF-α secreted by HSV-1 stimulated pDC and monocytes [48]. These data indicate that HSV-1 is a suitable stimulus to convert pDC from a tolerogenic to a fully-activated stage.

To investigate the effects of HSV-1-stimulated pDC on malignant melanoma, we co-cultivated pDC and melanoma cell lines in the presence of the replication-deficient HSV-1 *d*106S strain [49]. Notably, strong cytotoxic activities of pDC were observed, which were directed against ten of eleven melanoma cell lines, resulting in apoptotic and necrotic tumor cell death (Figure 1). These oncolytic effects, similarly noted in co-cultures with three leukemic cell lines, were dependent on type I IFN production and on viral mRNA transcription, as evident from increased melanoma cell viability in the presence of neutralizing antibodies against the IFN-α receptor and UV-irradiation of the viral stock, respectively. Notably, pDC were significantly more cytotoxic than NK cells using the same experimental conditions [49]. 

These data indicate that pDC can efficiently be activated by HSV-1 and subsequently exert anti-tumor activities in co-cultures. Stimulation of pDC with HSV-1 prior to co-culture significantly reduced the cytotoxic activity, which demonstrated that the infection of melanoma cell cultures by HSV-1 contributed to the oncolytic effect of pDC.

### 2.3. Onco-Lysis

Viruses do not only activate pDC but exert additional oncolytic effects (Figure 1). These effects, which have been observed using many different viruses in multiple tumor models in vitro and in vivo [11], rely on two intertwined concepts: (i) Viruses infect and directly lyse the tumor cells which subsequently release tumor-specific antigens; and (ii) these molecules are captured by antigen presenting cells and induce respective cytotoxic T-cells. Consequently, the oncolytic activities of viruses combine direct viral effects with indirect effects on innate and adaptive immune responses, the latter mostly triggered by DC cross-presentation of tumor-specific antigens. 

To enhance cross-presentation, oncolytic HSV-1 Talimogene Laherparepvec (T-VEC) was equipped with the gene for granulocyte-macrophage colony-stimulating factor (GM-CSF), which drives the recruitment and influx of antigen-presenting cells into the tumor [50]. This concept has been validated in phase II and phase III clinical trials, in which significant anti-melanoma effects were observed in hundreds of patients with local injections of T-VEC into the tumor compared to administration of GM-CSF only [51,52]. The significant regression of injected, but to some extent also of distant lesions [53,54], led to the approval of the first oncolytic herpes virus for the treatment of unresectable stage IIIB/C and IVM1a melanoma without bone, brain, lung, or other visceral metastasis by US and European authorities in 2015 and 2016, respectively. 

In further studies, T-VEC showed not only the shrinking of injected lesions but induced the regression of cutaneous, lymphatic, and visceral lesions, although not as efficiently as observed in directly inoculated lesions [55]. Swelling of the injected lesion prior to response did not affect clinical effectiveness [55]. The anti-tumor effects were attributed to virus replication in injected melanoma lesions and the induction of systemic immune responses resulting from tumor cell lysis with subsequent cross-presentation of tumor-specific antigens. 

### 2.4. More or Less Virulence, That Is the Question

In contrast to wild type HSV-1, T-VEC is an attenuated strain resulting from deletion of the γ34.5 gene, which is required for viral neurotoxicity via neutralization of protein kinase R (PKR) activity by inhibiting phosphorylation of eIF-2α [56], and from inactivation of the ICP47 gene, which counteracts the activity of the transporter associated with antigen processing (TAP) [57,58]. Despite these attenuating modifications, T-VEC is fully replicative in tumor cells. The ability to replicate is advantageous in oncolytic tumor therapies because progeny virus will spread to cells which have not been infected in the first round. On the downside, replication in the malignant lesion may provoke a strong inflammatory reaction, which may be detrimental, e.g., for brain tumors. In this respect, significant anti-tumor effects as well as immune-stimulating properties were observed with the replicative T-VEC strain in nude Balb/c mice [59].

Interestingly, new generations of oncolytic HSV-1 are designed to exhibit less virulence compared to T-VEC via deletions of further viral genes. In this respect, G47Δ and JX-594 have insertions in the immediate early gene ICP6 and thymidine kinase, respectively [60]. Another herpes virus, G207, which is currently evaluated in phase I studies for the treatment of pediatric brain tumors, has a deletion in the viral ribonucleotide reductase UL39 [61].

The HSV-1 *d*106S strain is completely replication-deficient due to deletion or inactivation of immediate early genes ICP4, ICP22, and ICP27 (in addition to ICP47) [62]. ICP4 and ICP22 are major viral transcription factors [63], and ICP27 blocks interferon production via downregulating of STAT1 phosphorylation [64], which leaves ICP0 and ICP6 as the only genes to be expressed after infection. Infectious virions are produced in a complementing cell line, which substitutes for the lack of deleted genes. Though replication-deficient, HSV-1 *d*106S has shown a strong oncolytic activity—in combination with pDC, in particular [49]. Notably, this activity was significantly reduced with UV irradiation, suggesting that efficient oncolysis by HSV-1 requires infection of susceptible tumor cells, but not completion of the replication cycle.

Interestingly, the oncolytic activity of the replication-deficient HSV-1 *d*106S—in contrast to most replication-competent viruses—was not decreased but increased in the presence of pDC-derived type I interferon production [49] (Figure 1). Usually, type I IFNs block viral replication, resulting in reduced anti-tumor effects of replicative viruses. In contrast, type I IFN production secreted by stimulated pDC enhanced the oncolytic activity of the non-replicative HSV-1 *d*106S. Obviously, replication-competent viruses are more susceptible to antiviral effects of IFNs compared to non-replicating viruses, which affects the anti-tumor effects.

In this respect, replication-deficient viruses may be most effective in tumors in which pDC are already present prior to infection. The downside of replication-deficient viruses may be that a higher multiplicity of infection is required for oncolytic activity, because the virus does not self-amplify within the tumor. In addition, it is still unclear whether the type of cell death is similar between replication-competent and replication-deficient herpes viruses, i.e., whether tumor cells infected with the latter die silently or stir an inflammatory microenvironment and turn a cold tumor into a hot zone.

### 2.5. Cross-Presentation vs. Direct Presentation

The long-term effects of oncolytic viruses are attributed to cross-presentation of tumor antigens released from dying cells, resulting in induction of cytotoxic T-cells (Figure 1). Tumor-infiltrating lymphocytes (TILs) have an important role in anti-melanoma immunity because their presence was favorably correlated with the prognosis of patients [65]. The transfer of tumor antigen-pulsed monocyte-derived DCs prolonged the survival in particular in a subset of patients with induction of tumor-specific CD8+ T-cells [66]. TILs recognize a broad spectrum of melanoma-associated antigens like MelanA/MART-1, tyrosinase, gp100, NY-ESO1, MAGE-A1, and MAGE-A3 [67]. In a more physiological setting, tumor peptides are loaded onto MHC I via cross-presentation, which has not only been reported for cDC1 cells but also for pDC [68]. However, effector functions of TILs are frequently impaired by the immunosuppressive tumor microenvironment [69]. Consequently, checkpoint inhibitors blocking the immunosuppressive molecules CTLA-4, PD-1, and PD-L1 have become a major breakthrough in the therapy of metastasized melanoma [70]. These data point to the fact that induction of tumor antigen-specific T-cells remains a major challenge.

To elicit tumor-specific adaptive immune responses is also a crucial challenge for oncolytic viruses. Proof-of-concept comes from studies in which the effect of T-VEC was enhanced with the concomitant application of checkpoint inhibitors, which block one of the inhibitory molecules in the interaction between dendritic cells and T-cells. In this respect, the oncolytic effect was at least doubled when T-VEC was combined with a PD-1 inhibitor [71]. These promising results appear to be re-affirmed in a large ongoing phase III trial [72]. Hence, one of the main questions is how tumor antigen-specific T-cells can be induced more effectively. 

We hypothesized that the enhanced expression of a tumor-specific antigen in the context of an oncolytic herpes virus may favorably affect the process of cell killing. Since MelanA/MART-1 is the most frequent target of CD8+ T-cells responses against melanoma in vitro and in vivo [20,73], we constructed a HSV-1 *d*106S-based oncolytic virus, which expressed the melanoma-associated antigen MelanA/MART-1 in melanoma cells not naturally expressing this tumor antigen [74]. When these cells were cocultured with an HLA-matched MelanA-specific CD8+ T-cell clone, we observed the activation and degranulation of T-cells, which resulted in significantly enhanced cell death upon infection with the MelanA-encoding HSV-1 *d*106S (Figure 1). These data indicate that tumor-specific antigens can be re-expressed in tumor cells upon infection with a respectively modified oncolytic herpes virus, thereby provoking a CD8+ T-cell adaptive immune responses in addition to pDC innate immune responses.

### 2.6. Self- vs. Neo-Antigens

Another crucial question is whether self or neo-antigens should be preferentially used to induce T-cell-mediated immunity. Recently, two seminal papers showed that neo-antigens served as efficient targets not only to induce in particular CD4+ T-cell and, to a lesser extent, CD8+ T-cell responses but also to elicit clinical responses in patients with malignant melanoma, namely reduction of metastasis [75,76]. Both studies investigated neo-antigens in a personalized vaccine approach, either via subcutaneous injection of adjuvanted peptides [76] or via intralymphatic administration of RNA synthesized from minigenes of tumor-specific neo-epitopes [75]. Very recently, this concept has been confirmed in a clinical trial using a neo-peptide vaccination in glioblastoma patients [77]. These studies provided proof-of-concept that clinically relevant T-cell immunity, in particular of CD4+ T-cells, can be induced against neo-epitopes. Notably, T-cell immunity was also induced against tumor-associated self-antigens, although immune responses were significantly weaker than against neo-antigens. In these studies, severe adverse effects were not reported.

So far, it is unclear whether the expression of tumor antigens or epitopes in the context of oncolytic viruses will actually contribute to the induction of tumor antigen-specific T-cells and may skew the immune reaction into a more pronounced CD8+ T-cell response. We have observed expression of HSV-1 *d*106S-encoded GFP upon infection of CD11c+ cells and macrophages, but we were so far not able to detect HSV-1 *d*106S-MelanA expression in antigen-presenting cells [74]. Therefore, this important point needs to be evaluated in further studies. The advantage of incorporating tumor-associated self-antigens or self-epitopes into an oncolytic herpes virus would be that a respectively designed oncolytic virus was suitable for a much larger patient cohort, while the use of neo-antigens or neo-epitopes would require a personalized design. 

HSV-1 *d*106S shows prolonged transgene expression and efficient presentation on MHC-I and MHC-II [78] and has been used in the past to induce humoral and cellular immune responses against viruses. In this respect, it induced responses against ß-galactosidase [78,79] and West Nile virus [80] in naïve and HSV-immune mice. In addition, HSV-1 *d*106S was used as vaccine to elicit both humoral and cellular immune responses against lentiviruses in the macaque model, which contributed to the control of virus replication [81]. Thus, HSV-1 *d*106S is, in principle, suited to be used as vaccine vector, although these antigens were of viral origin and thus “neo.” It remains to be investigated whether the expression of “self” in the context of an oncolytic virus is strong enough to break the immunological tolerance. If yes, it has to be studied whether the induced autoimmunity affects only the tumor—as intended—or other tissues as well. 

### 2.7. Designer Viruses

Besides expression of tumor-related antigens or epitopes (Figure 2), oncolytic herpes viruses can incorporate genes which contribute to tumor control in several respects. The most prominent example is T-VEC, which encodes GM-CSF to recruit immune cells to injected lesions [50]. Another approach is the expression of murine IL-12 in the HSV-1 context, which aims at enhancing adaptive immune responses, as has been confirmed in murine xenograft models [61]. Based on the success of this approach, two related oncolytic HSV-1 strains, which are currently evaluated in phase I studies of pediatric and adult brain tumors, were designed to express human IL-12 [82,83]. IL-12 was also incorporated into a fully-virulent HSV-1 strain, which was targeted to human epidermal growth factor receptor (HER) 2-expressing tumor cells, inducing a systemic immunotherapeutic vaccine response in a murine model [84]. Serious toxic side effects of IL-12 in tumor immunotherapies may be reduced using a variant of this molecule with deleted signal peptide [85].

Besides expressing growth factors or interleukins (Figure 2), oncolytic herpes viruses can also incorporate genes which turn the tumor environment into an uncomfortable place [86]. In this respect, herpes virus strains which code for enzymes to remove the extracellular matrix, thereby acting as physical barrier to limit viral spread, were generated and tested in orthotopic murine xenograft models [87,88]. Further approaches include incorporation of genes which limit the blood supply of tumors [89] and which express cell death molecules like TRAIL [90]. A very recent approach described cloning of the gene for the antibody against murine programmed cell death 1 (PD-1) into the oncolytic vaccinia virus, causing a massive infiltration of immune cells in a syngeneic murine tumor model [91]. A similar approach aiming at inducing high local immune checkpoint inhibitor concentrations may be realized with oncolytic herpes viruses. 

The incorporation of new genetic information into oncolytic HSV-1 has been laborious because HSV-1 is a large virus comprising more than 150,000 base pairs and does not support simple cloning. Therefore, genes of interest are usually cloned into a transfer plasmid and incorporated into the virus using homologous recombination [62]. However, this approach may not be successful on the first attempt and requires time-consuming plaque-purification of clonal viruses. The modification of oncolytic HSV-1 will profit much from the BAC technology, which was first established for cloning of cytomegalovirus [92] and has proven to be very useful in this context. This method will facilitate quick incorporation and exchange of genes within HSV-1 [86,93]. Roughly one fifth of the herpes viral genome is non-essential and therefore provides ample opportunities to incorporate genes coding for tumor antigens and/or chemo- and cytokines [93]. Future promising options may come from modifying the HSV-1 genome by CRISPR-Cas9 technology or from generating virus-like particles using synthetic genes.

## 3. Prospects of pDC in Enhancing Anti-Tumoral Immunity of Designed Oncolytic Herpes Viruses

As outlined above, pDC provide ample opportunities to enhance cytotoxic effects of oncolytic HSV-1:Secretion of anti-neoplastic type I IFNsActivation of NK cells via type I IFNsExertion of direct oncolytic activity by pDCAmplification of cytotoxic activity of oncolytic HSV-1Contribution to cross-presentation of tumor-associated antigens by DCInduction of tumor antigen-specific CD4+ and CD8+ T-cells

The genetic engineering of oncolytic HSV-1 contributes to several of these activities. The expression of cytokines and growth factors will enhance influx of immune cells into the injected lesions and thus increase cross-presentation of tumor-related antigens released by dying tumor cells. The expression of death molecules and anti-angiogenic factors will amplify viral cytotoxicity and oncolytic effects exerted by stimulated pDC. Incorporation of tumor-related self-antigens into oncolytic herpes viruses will result in re-expression of these antigens in tumor cells, which have escaped the immune response, although MHC I downregulation on tumor cells may impair presentation of peptides from HSV-encoded tumor antigens. Upon infection, these cells will be killed by tumor antigen-specific T-cells. Expression of self- and neo-antigens may contribute to the (cross-) presentation of these antigens via DC (and possibly via pDC), which will induce expansion and functionality of respective tumor antigen-specific CD4+ and CD8+ T-cells. The latter effects will specifically be promoted by high local concentrations of checkpoint inhibitors.

pDC may amplify cytotoxic effects of oncolytic herpes viruses in particular in those tumors, in which they are already present prior to injection of the virus. It may not be by chance that the first tumors to be successfully evaluated for oncolytic HSV-1 therapy are malignant melanoma and head and neck cancers. Malignant melanoma is known for being an immunogenic type of cancer, leading the phalanx of tumors with the largest expression of neo-antigens [94]. Melanoma is also known for being infiltrated and surrounded by pDC, which has also been reported for head and neck carcinoma. Specifically, the presence of pDC may be crucial when non-replicating viruses are used for oncolytic therapy, because the viruses will only be present for a short time within the tumor. Another favorable aspect is the susceptibility of certain tumors, e.g., malignant melanoma, to the antiproliferative, antiangiogenic, and immunostimulatory effects of type I IFNs, which is exploited clinically in adjuvant IFN-α2 therapy for certain stages of melanoma [95]. 

An additional important point is the cooperation of pDC and classical DC in orchestrating adaptive anti-tumor T-cell responses. In this respect, TLR9-stimulated pDC were reported to induce tumor antigen cross-presentation by conventional DC to CD8+ T-cells in the murine model [96]. HSV-1 is known to induce pDC activation via TLR9-dependent and TLR9-independent pathways [97]. Thus, HSV-1 combines strong activation of pDC with the virus-mediated oncolysis of tumor cells, which promotes the release of pathogen- and damage-associated molecular patterns (PAMPs, DAMPs), as well as tumor-associated antigens. Designing oncolytic HSV-1 to directly express tumor-related self- and neo-antigens may contribute to a vaccine-like approach resulting in a more efficient induction of cytotoxic T-cell responses. It may be helpful to express tumor-related antigens or epitopes by less virulent viruses, because cells which rapidly succumb to oncolytic effects may not efficiently express the genes of interest [74].

The effect of oncolytic viral vaccines may be more pronounced in the context of checkpoint inhibitors. In this respect, the effect of T-VEC at least doubled with parallel administration of ipilimumab and pembrolizumab [71,98]. Considering these data, the concept of vaccination has recently been revitalized in the context of checkpoint inhibitors. In this context, B cells and antibodies against tumor-related antigens may play a role as biomarkers for response and survival [99]. Finally, it may be worth to test oncolytic viruses in neoadjuvant vaccination approaches, which may generate an efficient T-cell response prior to removal of the tumor and thus may prevent or delay tumor relapse after surgery.

## 4. Conclusions

Oncolytic herpes viruses provide a unique opportunity to treat and to vaccinate against certain tumors. These effects are amplified by pDC, which surround and infiltrate certain types of cancer. HSV-1 can incorporate large amounts of foreign DNA and thus be adjusted to the individual needs of many tumor entities. Besides incorporating genes which turn the tumor into a hot zone, e.g., chemokines and growth factors, HSV-1 can be engineered to express self- or neo-antigens, which may serve as therapeutic or neoadjuvant tumor vaccine in particular in the presence of immune checkpoint inhibitors. Oncolytic HSV-1 can thus enhance activation and expansion of tumor antigen-specific T-cells as well as the access of these cells to the tumor. The prospects of a HSV-1-encoded tumor vaccine are particularly promising in the light of combination tumor immunotherapies [9]. Future prospects will come from current clinical trials evaluating oncolytic HSV in different tumor entities (Table 1).

## Figures and Tables

**Figure 1 cancers-11-00651-f001:**
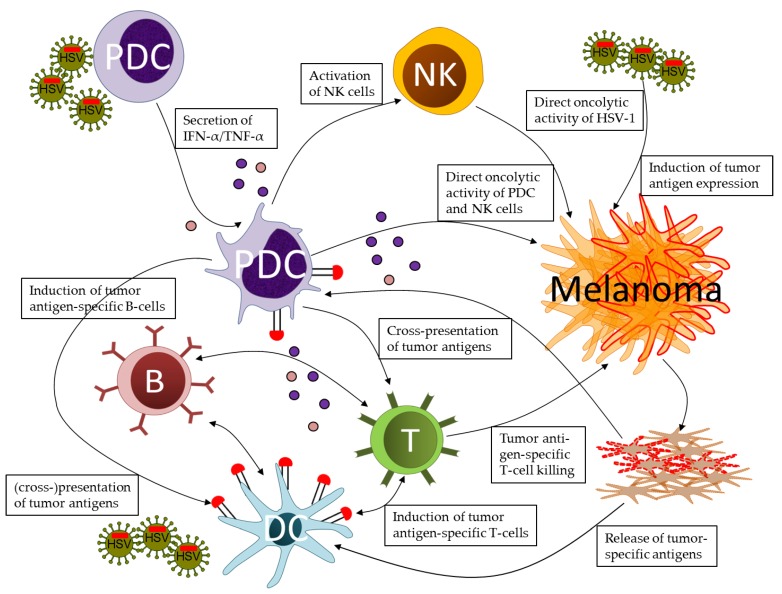
Prospects of plasmacytoid dendritic cells (pDC) in orchestrating innate and adaptive immune responses against melanoma in the context of oncolytic herpes simplex virus (HSV) infections. Upon stimulation by wild type, attenuated, or replication-deficient HSV-1, pDC secrete IFN-alpha and TNF-alpha, which activate natural killer (NK) cells. NK cells and activated pDC attack the tumor cells via soluble and cell-associated cytotoxic mechanisms. These effects contribute to melanoma cell death induced by infection with oncolytic HSV-1, either via lytic replication or induction of apoptotic and necrotic cell death. Dying tumor cells release melanoma-associated antigens, which are cross-presented to T-cells by classical dendritic cells (DC), mostly cDC1, and, at least in part, by pDC. The close cooperation of pDC and DC results in the induction of tumor antigen-specific CD4+ and CD8+ T-cells, which contribute to long-term control of tumor cells. Tumor escape from immune responses may result in loss of tumor antigen expression, which may be counteracted by oncolytic HSV encoding for these antigens. Oncolytic HSV-1 expressing either self-antigens or neo-antigens may further serve as tumor vaccines in adjuvant and neo-adjuvant applications.

**Figure 2 cancers-11-00651-f002:**
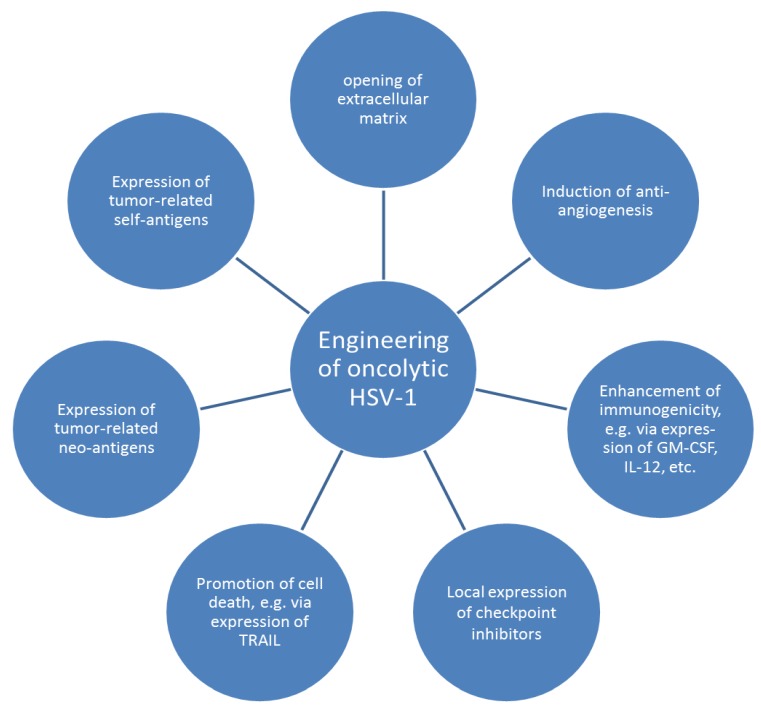
Prospects of optimizing anti-tumor activities via genetic engineering of oncolytic herpes simplex viruses 1 (HSV-1). HSV-1 can be designed to express proteins which modify the tumor environment, e.g., via restricting blood supply of the tumor, reducing barriers for efficient viral spread, enhancing influx of immune cells through production of pro-inflammatory cytokines, and/or contributing to local increase of immune cell activity. Engineered HSV-1 may also promote tumor cell lysis via expression of apoptosis-inducing ligands or stimulate different cells of the adaptive immune system as vaccine vector presenting neo- or self-antigens. Adapted according to [86].

**Table 1 cancers-11-00651-t001:** Current clinical trials using oncolytic HSV (as listed on ClinicalTrials.gov, accessed April 2019).

HSV	Study	Identifier	Phase
OH2	Phase I Study of OH2 Injection, an Oncolytic Type 2 Herpes Simplex Virus Expressing Granulocyte Macrophage Colony-Stimulating Factor, in Malignant Solid Tumors	NCT03866525	1
rRp450	rRp450-Phase I Trial in Liver Metastases and Primary Liver Tumors	NCT01071941	1
Orien X010	Recombinant Human GM-CSF Herpes Simplex Virus Injection (OrienX010), Standard Injection in Tumor, Treatment Scheme Failed, M1c IV Period, Malignant Melanoma Spread to the Liver, Open I-c Phase of Clinical Trial	NCT03048253	1c
M032	A Phase 1 Study of M032 (NSC 733972), a Genetically Engineered HSV-1 Expressing IL-12, in Patients With Recurrent/Progressive Glioblastoma Multiforme, Anaplastic Astrocytoma, or Gliosarcoma	NCT02062827	1
C134	A Phase I Trial of IRS-1 HSV C134 Administered Intratumorally in Patients With Recurrent Malignant Glioma	NCT03657576	1
G207	Phase 1 Trial of Engineered HSV G207 in Children With Recurrent or Refractory Cerebellar Brain Tumors	NCT03911388	1
G207	Phase I Clinical Trial of HSV G207 Alone or With a Single Radiation Dose in Children With Recurrent Supratentorial Brain Tumors	NCT02457845	1
T-VEC	A Phase I, Open Label, Single Arm, Single Centre Study to Evaluate Mechanism of Action of Talimogene Laherparepvec (T-VEC) in Locally Advanced Non-melanoma Skin Cancer	NCT03458117	1
T-VEC	A Phase 1/2 Study of Talimogene Laherparepvec in Combination With Neoadjuvant Chemotherapy in Triple Negative Breast Cancer	NCT02779855	1/2
T-VEC	A Phase II Study Using Talimogene Laherparepvec as a Single Agent for Inflammatory Breast Cancer (IBC) or Non-IBC Patients With Inoperable Local Recurrence	NCT02658812	2
T-VEC	A Phase II Study of Talimogene Laherparepvec Followed by Talimogene Laherparepvec + Nivolumab in Refractory T Cell and NK Cell Lymphomas, Cutaneous Squamous Cell Carcinoma, Merkel Cell Carcinoma, and Other Rare Skin Tumors	NCT02978625	2
T-VEC	A Phase 1b Study of Talimogene Laherparepvec (T-VEC) in Combination With Paclitaxel or Endocrine Therapy in Patients With Metastatic, Unresectable, or Locoregionally Recurrent HER2-Negative Breast Cancer With Evidence of Injectable Disease in the Locoregional Area	NCT03554044	1b
T-VEC	A Phase II Study of Combining Talimogene Laherparepvec T-VEC (NSC-785349) and MK-3475 (Pembrolizumab) (NSC-776864) in Patients With Advanced Melanoma Who Have Progressed on Anti-PD1/L1 Based Therapy	NCT02965716	2
T-VEC	A Phase I Study of Talimogene Laherparepvec (TALIMOGENE LAHERPAREPVEC) With Neoadjuvant Chemotherapy and Radiation in Adenocarcinoma of the Rectum	NCT03300544	1

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
