# Peer review of "Prospect of Plasmacytoid Dendritic Cells in Enhancing Anti-Tumor Immunity of Oncolytic Herpes Viruses"

_cancers, 2019, doi:10.3390/cancers11050651_

Round 1

Reviewer 1 Report

Schuster et al.in their manuscript review the role of plasmacytoid dendritic cells (PDC) in tumor immune-surveillance in context of oncolytic virus therapies particularly with herpes simplex virus-1 (HSV-1). PDCs have been shown to be presented in certain tumors and their presence is correlated with bad prognosis of patients. However, if activated, PDCs get antitumor properties by direct cytotoxic activity or by supporting adaptive anti-tumor responses. PDCs are also key components in anti-viral immunity by production of type I interferons. Therefore, PDCs might play a dual role in oncolytic virotherapies of cancer. Thus the topic of this review is very timely and highly relevant.   

Major points:

Oncolytic viruses represent an emerging tool in cancer immunotherapies, but a general overview on HSV would be helpful for readers experienced in cancer immunobiology but not having virologic backgrounds.

The title implies a focus on the role of PDCs in oncolytic herpes virus therapies. Nevertheless, majority of chapters (2.3., 2.4., 2.5., 2.6. and 2.7.) only discuss mechanisms of HVS oncolytic virus therapies.

A chapter highlighting the involvement of PDCs in oncolytic virotherapies other than HSV would be interesting with examples from both replicating and replication-deficient oncolytic viruses.  

As stated correctly by the authors, IL-12 is a very potent adjuvant for cancer immunotherapies, however serious toxicity issues should also be mentioned. 

Author Response

Reviewer 1:

Schuster et al.in their manuscript review the role of plasmacytoid dendritic cells (PDC) in tumor immune-surveillance in context of oncolytic virus therapies particularly with herpes simplex virus-1 (HSV-1). PDCs have been shown to be presented in certain tumors and their presence is correlated with bad prognosis of patients. However, if activated, PDCs get antitumor properties by direct cytotoxic activity or by supporting adaptive anti-tumor responses. PDCs are also key components in anti-viral immunity by production of type I interferons. Therefore, PDCs might play a dual role in oncolytic virotherapies of cancer. Thus the topic of this review is very timely and highly relevant.   

Major points:

Oncolytic viruses represent an emerging tool in cancer immunotherapies, but a general overview on HSV would be helpful for readers experienced in cancer immunobiology but not having virologic backgrounds.

Authors: We agree with the reviewer and have included more detailed information about HSV-1 into section 2.2. (“The viral wake-up call”).

The title implies a focus on the role of PDCs in oncolytic herpes virus therapies. Nevertheless, majority of chapters (2.3., 2.4., 2.5., 2.6. and 2.7.) only discuss mechanisms of HVS oncolytic virus therapies.

Authors: The aim of our manuscript was to focus on the interaction of oncolytic herpes viruses and pDC. To address the reviewer’s concerns that our manuscript may be a little on the short side for pDC, we added further references for the heterogeneity of pDC (section 1.1., Introduction; PMID 29925996), for the impaired activation of pDC via TLR9-mediated stimuli (section 2.1., “The yin and yang of pDC within tumors”; PMID 23722543), for the interaction of PDC with viruses other than HSV-1 (section 2.2, “The viral wake-up call”; PMID 28197384, PMID 23339127, PMID 31011627), and for the ability of pDC to cross-present antigens (section 2.5; “Cross-presentation vs. direct presentation”; PMID 23339127). 

A chapter highlighting the involvement of PDCs in oncolytic virotherapies other than HSV would be interesting with examples from both replicating and replication-deficient oncolytic viruses.

Authors: This is an interesting question, but actually there is not a lot known for pDCs and other oncolytic virotherapies besides HSV-1. We had already included most of these papers in the manuscript: the Plumas group used inactivated influenzaviruses or flu-like virosomes, the Figdor group investigated a series of live-attenuated and inactivated viral vaccines, in particular tick-borne encephalitis virus. We added two additional papers, which describe TRAIL-mediated cytotoxicity and enhanced tumor antigen cross-presentation by human pDC upon stimulation with live-attenuated measles virus vaccine (PMID 28197384, PMID 23339127) to section 2.2. (“The viral wake-up call”), and another paper describing recruitment of pDC to mouse melanoma lesions upon exposure to human ß-defensin-expressing vaccinia virus (PMID 31011627).

As stated correctly by the authors, IL-12 is a very potent adjuvant for cancer immunotherapies, however serious toxicity issues should also be mentioned. 

Authors: We agree that serious side effects have been reported for IL-12 in cancer immunotherapies. We now mention these toxicity issues in chapter 2.7. (“Designer viruses”), referencing a paper by Wang et al. (Nat Commun 8:1395), who reported that the toxicity is significantly reduced with an oncolytic adenovirus expressing an IL-12 variant depleted for the signal peptide.

Reviewer 2 Report

Schuster et al reviews the potential of HSV oncolytic virus mediated anti tumor immunity and the potential contribution of the pDC compartment. Generally the review highlighted different potential possibility of the HSV system for developing a tool for vaccination. 

There are few comments and it will be great if the authors can  incorporate the informations to the review. 

I think the generally used abbreviation s for plasmacytoid DCs are pDC  than PDCs.

Line 37. PDCs were identified by other teams for and including their ability to produce IFNa PMID: 2532619 and PMID: 8102389. Eventhough they may not be well defined as pDCs.

Line 53: pDC activation with TLRs: There are specific reports on the TLR9 malfunction on on tumor infiltrating pDCs (PMID: 23722543)

How important is the role of pDCs in an oncolytic virus infection. The virus infection is specific to the tumor cells and the direct lysis helps other APCs to take up the antigen and cross present to the T cells. A productive infection of pDCs is not reported with general HSV strains and how important is a productive infection of pDCs to generate an antitumor response. There are recent reports on the heterogeneity of the cell subsets identified as pDCs and some of the upcoming studies clearly shows that virus does not infect pDCs https://www.biorxiv.org/content/10.1101/578377v1). The infection is more directed towards the CDP population present in pDCs.  Even the group has data on HSV pseudo typed HIVs fails to infect pDCs.  I think the earlier paper was developed based on the concept of HSV infection of pDCs.  How does the authors explain the discrepancies and how important is pDC- HSV axis in an in vivo situation?  

Line 220 :  Self vs non-self: The concept of introducing tumor associated antigens or neo antigens (at least shared neo antigens like KRAS or antigens associated with micro satellite instability) are interesting. There are other draw backs including HLA restrictions and other general problems like a skewed CD4 response than CD8 response which is critical for antitumor immune response. What additional advantage the authors envisage neo antigen or TAA expressing HSV compared to the injection of RNA or peptide to the patients (much easier compared to develop an personalized HSV in GMP setting)? Authors also propose the addition of cytokines to boost the antitumor immune response. It will be great to enlighten on the potential size of inserts which can be incorporated to HSV system and what is the size limit for co-delivering a cytokine gene and TAA or neoantigens?  The oncolytic viruses have a restricted infection and expression of these molecules, specifically in tumor. Tumors in general down regulate the MHC I and hence there is less chance of direct presentation of  HSV encoded antigens.  So whether these approaches have any genuine application or mere academicals interest ?

Role of pDCs in tumor regression: Mouse models clearly demonstrated the role of cDC1 on antitumor immunity. Even the presence of pDCs does not do anything critical for antitumor immune response in most of the mouse models ( Studies from Ken Murphy’ s team are example , including the BATF3 model to recent WDFY4). Another critical study demonstrate the role of cDC1 Spranger et al. 2017 and even they show the effective response for check point inhibitors, cDC1 is critical. All these mouse models have an impaired pDC compartment and still fail to induce a strong anti-tumor response.

A number of recent reports in human studies shows a similar observation and critical role of cDC1 (Roberts  et al, Cancer Cell. 2016 Aug 8;30(2):324-336 , Broz et. al , Cancer Cell. 2014 Nov 10;26(5):638-52, Böttcher  et. al, Cell. 2018 Feb 22;172(5):1022-1037, Barry et. al, Nat Med. 2018 Aug;24(8):1178-1191. , Kyi et. al, Clin Cancer Res. 2018 Oct 15;24(20):4937-4948, Michea et. al, Nat Immunol. (2018) 19:885–97). All these tumors also have pDC and cDC2 compartment and even though the cDC1 is ten times less than pDCs or cDC2, still play a critical role in human antitumor response.   How authors explain these facts and in context of a pDC axis for generating antitumor tumor immunity.

I think the authors have to consider the critical role played by other DC subsets and pDCs role may be mainly restricted to the type I IFN production and activate the effector cells or improve the  even cross presentation than playing a lead role in antitumor immune response.

Please add a table listing the current clinical trials conducted using HSVs against various malignancies

Author Response

Reviewer 2:

Schuster et al reviews the potential of HSV oncolytic virus mediated anti tumor immunity and the potential contribution of the pDC compartment. Generally the review highlighted different potential possibility of the HSV system for developing a tool for vaccination. 

There are few comments and it will be great if the authors can incorporate the informations to the review. 

I think the generally used abbreviations for plasmacytoid DCs are pDC than PDCs.

Authors: In the beginning of the PDC era, the abbreviation pDC was also used for precursor DC, which may develop into PDC, but also into myeloid DC. However, we agree that pDC is more common nowadays and therefore changed the abbreviation in the manuscript.

Line 37. PDCs were identified by other teams for and including their ability to produce IFNa PMID: 2532619 and PMID: 8102389. Eventhough they may not be well defined as pDCs.

Authors: We thank the reviewer for highlighting these early papers to us. We have included these references into the introduction.

Line 53: pDC activation with TLRs: There are specific reports on the TLR9 malfunction on on tumor infiltrating pDCs (PMID: 23722543)

Authors: We thank the reviewer for contributing this important article and have included the reference into the first paragraph of section 2.1 (“The yin and yang of pDC within tumors”).

How important is the role of pDCs in an oncolytic virus infection. The virus infection is specific to the tumor cells and the direct lysis helps other APCs to take up the antigen and cross present to the T cells. A productive infection of pDCs is not reported with general HSV strains and how important is a productive infection of pDCs to generate an antitumor response. There are recent reports on the heterogeneity of the cell subsets identified as pDCs and some of the upcoming studies clearly shows that virus does not infect pDCs https://www.biorxiv.org/content/10.1101/578377v1). The infection is more directed towards the CDP population present in pDCs.  Even the group has data on HSV pseudo typed HIVs fails to infect pDCs.  I think the earlier paper was developed based on the concept of HSV infection of pDCs.  How does the authors explain the discrepancies and how important is pDC- HSV axis in an in vivo situation?  

Authors: The reviewer wants to know how important we consider infection of pDC by HSV-1, arguing that only the SIGLEC-1 expressing precursor DC are infected by HIV-1. In the first place, we agree that pDC are a heterogenous cell population and included the recent paper by Rodrigues et al. (PMID 29925996) describing that pDC are composed of a majority of “conventional” pDC and a minority of “pDC-like cells” into the Introduction. According to the authors, the latter excel in antigen processing and presentation and may thus contribute to inducing antitumor responses in vivo.

Concerning HSV-1 infection of pDC, we and others have shown that the virus enters these cells, however, transcription of viral genes and productive infection does not occur (Donaghy et al., 2009, PMID 19073735; Schuster et al., 2010, PMID 19824924). Currently, it remains unclear whether infection of pDC and subsequent expression of transgenes under the control of (human) promotors may contribute to tumor regression. On the one hand, cytotoxic effects of PDC are independent of productive infection of these cells. On the other hand, data support cross-presentation by pDC (Guillerme et al., 2013, PMID 23339127). We consider these two effects in addition to secretion of type I interferons and other pro-inflammatory cytokines as the major anti-tumor effects of pDC. To address the reviewer’s point, we modified Figure 1 accordingly to shift the focus from infection of pDC to cross-presentation by these cells.  

Line 220:  Self vs non-self: The concept of introducing tumor associated antigens or neo antigens (at least shared neo antigens like KRAS or antigens associated with micro satellite instability) are interesting. There are other draw backs including HLA restrictions and other general problems like a skewed CD4 response than CD8 response which is critical for antitumor immune response. What additional advantage the authors envisage neo antigen or TAA expressing HSV compared to the injection of RNA or peptide to the patients (much easier compared to develop an personalized HSV in GMP setting)? Authors also propose the addition of cytokines to boost the antitumor immune response. It will be great to enlighten on the potential size of inserts which can be incorporated to HSV system and what is the size limit for co-delivering a cytokine gene and TAA or neoantigens?  The oncolytic viruses have a restricted infection and expression of these molecules, specifically in tumor. Tumors in general down regulate the MHC I and hence there is less chance of direct presentation of  HSV encoded antigens.  So whether these approaches have any genuine application or mere academicals interest ?

Authors: As the reviewer says, peptides or RNA are more suitable for a personalized tumor approach compared to incorporation of individual neo-antigens into a viral backbone. The pronounced CD4+ T-cell response following peptide or RNA vaccination, however,  may argue that the ideal way of stimulating CD8+ T-cell responses still has to be defined. The hope is that the expression of a tumor antigen in the viral context may skew the immune reaction into a more pronounced CD8+ T-cell response, which has to be analyzed in further studies. We expressed these ideas now more clearly in section 2.6. (“Self vs. neo-antigens”).

In addition, the reviewer is interested in the size limit for insertions in the HSV system. The BAC system by Bailer et al. (cited in the manuscript) describes that roughly 30 kB of the 152 kB herpes viral genome are non-essential, which provides ample opportunities to incorporate genes coding for tumor antigens and/or chemo- and cytokines. This information is now included into section 2.7. (“Designer viruses”).

The reviewer argues that downregulation of MHC I expression on tumors may impair presentation of tumor-associated peptides from HSV-encoded antigens. This may very well be possible, however should similarly be true for the RNA-based approach, which has proven to be effective in vivo (Sahin et al. 2017). In response to the reviewer’s concern, we included this point into section 3 (“Prospects of pDC in enhancing anti-tumoral immunity of designed oncolytic herpes viruses”).

Role of pDCs in tumor regression: Mouse models clearly demonstrated the role of cDC1 on antitumor immunity. Even the presence of pDCs does not do anything critical for antitumor immune response in most of the mouse models ( Studies from Ken Murphy’ s team are example , including the BATF3 model to recent WDFY4). Another critical study demonstrate the role of cDC1 Spranger et al. 2017 and even they show the effective response for check point inhibitors, cDC1 is critical. All these mouse models have an impaired pDC compartment and still fail to induce a strong anti-tumor response.

A number of recent reports in human studies shows a similar observation and critical role of cDC1 (Roberts  et al, Cancer Cell. 2016 Aug 8;30(2):324-336 , Broz et. al , Cancer Cell. 2014 Nov 10;26(5):638-52, Böttcher  et. al, Cell. 2018 Feb 22;172(5):1022-1037, Barry et. al, Nat Med. 2018 Aug;24(8):1178-1191. , Kyi et. al, Clin Cancer Res. 2018 Oct 15;24(20):4937-4948, Michea et. al, Nat Immunol. (2018) 19:885–97). All these tumors also have pDC and cDC2 compartment and even though the cDC1 is ten times less than pDCs or cDC2, still play a critical role in human antitumor response. How authors explain these facts and in context of a pDC axis for generating antitumor tumor immunity.

I think the authors have to consider the critical role played by other DC subsets and pDCs role may be mainly restricted to the type I IFN production and activate the effector cells or improve the even cross presentation than playing a lead role in antitumor immune response.

Authors: The reviewer points out the important role of cDC1 cells. Our manuscript certainly focused on the role of PDC in anti-tumor responses induced by oncolytic viruses. However, it should be noticed that we did not ignore myeloid DC, but described their important role on several occasions in the text as well as in Figure 1. We agree that we should have given more detail to cDC1 cells and added this information plus references to section 2.1. (“The yin and yang of pDC within tumors”).

In addition, the reviewer suggests that we overstated the role of pDC in anti-tumor immunity, in particular because mouse models do not see a major role of these cells. While TLR9 is expressed by murine pDC and mDC, TLR9 is expressed in human pDC only (Wagner et al., 2004, PMID 15207506). Therefore, we still think that pDC are particularly important for the treatment of human tumors using oncolytic HSV-1. We do not think that pDC are the only cell type to be important for tumor cell killing, but, as outlined in the text, activated pDC kill tumor cells, produce pro-inflammatory cytokines, stimulate NK cells, enhance cross-presentation by other DC, and thus are capable of triggering the anti-tumoral responses.   

Please add a table listing the current clinical trials conducted using HSVs against various malignancies

 Authors: We added a Table of current clinical trials using oncolytic HSV, as listed on ClinicalTrials.gov (accessed on April 30, 2019).

Round 2

Reviewer 1 Report

The authors improved the manuscript according to my concerns.

Reviewer 2 Report

The authors are answered the concerns and added additional information to improve the manuscript.